# Monte Carlo Simulations of Heat Deposition during Photothermal Skin Cancer Therapy Using Nanoparticles

**DOI:** 10.3390/biom9080343

**Published:** 2019-08-05

**Authors:** J. Charles G. Jeynes, Freddy Wordingham, Laura J. Moran, Alison Curnow, Tim J. Harries

**Affiliations:** 1Living Systems Institute, University of Exeter, EX4 4PU, UK; 2Physics and Astronomy, University of Exeter, EX4 4PU, UK; 3University of Exeter Medical School, University of Exeter, EX4 4PU, UK

**Keywords:** Monte Carlo simulations, photodynamic therapy, photothermal therapy, nanoparticles, gold nanorods, theranostics

## Abstract

Photothermal therapy using nanoparticles is a promising new approach for the treatment of cancer. The principle is to utilise plasmonic nanoparticle light interaction for efficient heat conversion. However, there are many hurdles to overcome before it can be accepted in clinical practice. One issue is a current poor characterization of the thermal dose that is distributed over the tumour region and the surrounding normal tissue. Here, we use Monte Carlo simulations of photon radiative transfer through tissue and subsequent heat diffusion calculations, to model the spatial thermal dose in a skin cancer model. We validate our heat rise simulations against experimental data from the literature and estimate the concentration of nanorods in the tumor that are associated with the heat rise. We use the cumulative equivalent minutes at 43 °C (CEM43) metric to analyse the percentage cell kill across the tumour and the surrounding normal tissue. Overall, we show that computer simulations of photothermal therapy are an invaluable tool to fully characterize thermal dose within tumour and normal tissue.

## 1. Introduction

Photothermal therapy is the application of light to induce hyperthermia in a target region of tissue, usually using plasmonic nanoparticles for enhanced localized heating (for a review see [1]). There are many reports of nanoparticle photothermal therapy, both in vivo and in vitro, using a wide variety of nanoparticle shapes, sizes and materials [1]. In this work, we investigate the theoretical thermal rise in gold nanorod infused tissue. Gold nanorods can be designed with an aspect ratio that is highly absorbing in the near infra-red light region due to a surface plasmon resonance effect [2]. Also, nanoparticles made of gold are likely to be non-toxic and so a more realistic proposition for regulatory bodies than other metal nanoparticle types.

Photothermal therapy is perhaps most useful for skin cancers as they generally lie within the range of infra-red light which has a penetration of about 4–5 mm into tissue [3]. Importantly, skin cancer is by far the most common form of cancer found in Caucasian populations worldwide, with one particular type—basal cell carcinoma (BCC)—presenting around 100,000 cases per year in the United Kingdom alone [4]. These cancers are rarely life threatening but left untreated can be locally invasive causing pain, bleeding and disfiguration. Further, a very small proportion can metastasize and lead to death, so it is essential that all cases be treated.

There are a number of treatment regimes for non-melanoma skin cancers and precancers (such as BCC, actinic keratosis and Bowen’s disease) including surgical excision, cryotherapy, radiotherapy and chemotherapeutic topical creams. However, an alternative treatment—photodynamic therapy (PDT)—has gained popularity due to ease of treatment, ability to treat multiple closely spaced tumours, repeatability without the development of resistance and excellent cosmetic outcomes without scar formation.

The principle of this form of PDT utilizes the interaction of visible red light (635 nm) with a photosensitizing drug (protoporphyrin IX (PpIX)). Protoporphyrin IX accumulates predominantly within tumour cells after application as a topical cream, and in the presence of molecular oxygen and red light, reactive oxygen species are produced which result in cell death [5,6].

However, PDT has limitations: (i) it is an oxygen-dependent process resulting in poor treatment efficacy with hypoxic tumours; (ii) PDT treatments generally use visible red light with limited penetration depth in tissue (about 3 mm); (iii) PpIX has a very small absorption coefficient with commercial PDT light sources making it an inefficient photosensitizer. Thus, there is a need for photosensitizing agents with absorption coefficients that closely match the excitatory wavelengths of near infrared (NIR) light sources, which are associated with deeper tissue penetration and furthermore do not rely on the generation of reactive oxygen species as a means for causing cell death.

To this end, nanoparticles have been researched intensively due to their tunable absorption coefficient depending on material properties, scale and aspect ratio [2], and thus many types of nanoparticle formulations exist with extinction coefficients in the NIR region. The principles of photothermal therapy using cancer models in mice have been demonstrated using a variety of nanoparticles, and gold nanorods [7]. Generally, these studies have either directly injected nanoparticles into tumours, or intravenously injected into the tail veins of mice and relied on the enhanced permeability and retention (EPR) of tumours for localized high nanoparticle concentrations. Of note, nanoparticles can also penetrate the skin after application in a cream, but penetration depth and associated spatial concentration is still a matter of debate [8,9,10].

Modelling is an invaluable tool to test the feasibility of treatment regimes with nanoparticles and to understand thermal dose distributions within tissue. Indeed, Monte Carlo modelling has successfully been employed to understand local dose distributions and fluorescence intensities arising from irradiated skin tumours treated with conventional PDT regimes [11,12,13,14]. There are few reports of in silico modelling of thermal dose from nanoparticles after irradiation with NIR light—recently a Geant4 based code was used to model photothermal enhancement from gold nanoparticles [15].

Here, we use our own custom-written Monte Carlo model—called ARCTORUS—based upon principles found within our astrophysical code TORUS (for a full review of TORUS see [16]), that simulates the processes of absorbance, scattering, refraction, and reflection of individual photon packets though optical media. We then use the heat transfer package ‘k-wave’ (www.k-wave.org) to simulate heat diffusion over the tissue. Our simulations fit well with experimental data and are particularly useful in dissecting heat rise throughout the tissue over time, estimating concentration of nanoparticles in tumours when this data is not experimentally available and assessing the likely percentage of cell death. Overall, this is important for predicting tumour control and damage to normal tissue.

## 2. Methods

A Monte Carlo radiative transfer code was developed in C++ to simulate the scattering and absorption processes present within the photothermal treatment process. The code calculates the spatial energy density profiles originating from an arbitrary light source incident upon a biological system. The system was modelled to geometrically characterize the arrangement of various optical materials spatially bounded by three-dimensional triangular meshes. These geometric objects are referred to as entities.

The presented simulation models a three-dimensional cuboid-shaped domain of 10 × 10 × 10 mm (x,y,z). A tumour entity was modelled as an oblate-spheroid 0.5 × 0.5 × 0.25 mm (x,y,z) and positioned 0.25 mm beneath the surface of a homogeneous scattering medium (flesh) entity. The flesh entity was a cube 6 × 6 × 6 mm (x,y,z) and was centrally suspended inside air which filled the domain. The entity dimensions were chosen such that boundary effects were significantly far removed from the volume of investigation (tumour and immediate volume). A light source with intensity of 1 W and a surface described by a circular triangle-fan (*n* = 16) mesh (radius 1 mm) was located at the top of the domain directly above the scattering medium. The optical properties of the tumour and scattering material were primarily characterized by the scattering (*μ*_s_) and absorption (*μ*_a_) coefficients and were left as free parameters.

The domain of the simulation was sub-divided uniformly into grid cells by partitioning the volume uniformly 201 times along each cartesian-axis. Each cell recorded the number of triangles permeating it, enabling the determination of the mesh body that a given point position fell within. Consequently, the physical properties of the simulation were determined primarily using the triangle-meshes of the entity objects.

One of the core functionalities of the code was to calculate the temperature distribution throughout the domain illuminated by photon sources (in this simulation a laser). Differences in photo-thermal heating between the tumour and flesh materials resulting from the laser illumination were measured. The core photon-loop of the code was based upon the radiative equilibrium routine as described in [16].

The total energy of the illuminating radiation from luminosity *L* over the specified duration of simulation Δ*t* was divided into *N* packets of energy defined by Equation (1):(1)$$\frac{\varepsilon }{\Delta t}=\frac{L}{N}$$

Each packet of energy may have any wavelength (λ) but the total energy of the photon packet is the same; photons packets of different wavelengths essentially just carry different numbers of photons. Each photon is emitted from a random location on a light source’s surface. The initial direction of the photon is normal to the surface at this point.

Following emission a photon packet travels forward, along its direction vector, a random distance:(2)$$l=\frac{-\ln\left(1-\xi \right)}{\mu }$$
where *ξ* is a uniform random deviate (0–1) and μ is the interaction coefficient of the current medium as experienced by the photon packet.

Each material within the simulation maintained a table specifying the optical scattering and absorption coefficients for photon wavelengths of interest to the simulation. The interaction coefficient for the medium was defined as:(3)$$\mu =\mu_{s}+\mu_{a}$$
and the single scattering albedo was defined as:(4)$$A\equiv \;\frac{\mu_{a}}{\mu }=\frac{\mu_{a}}{\mu_{a}+\mu_{s}}$$

After travelling distance l, the packet then interacted (scattered) within the medium. The packet’s statistical weight (w) was reduced to reflect the fraction of photons which would have been absorbed during the interaction process, leaving a remaining fraction of photons which would have been scattered away.
(5)$$w^{\prime }=w\cdot A$$

The deflection angle at which the packets scatter was determined by the scattering anisotropy factor (g) of the medium. The Henyey–Greenstein phase function was used to approximate scattering in tissue and is of the form:(6)$$P_{\theta }=\frac{1}{4\pi }\frac{1-g^{2}}{{\left(1+g^{2}-2g\cdot \cos\left(\theta \right)\right)}^{3/2}}$$
where $P_{\theta }$ is a probability density function and θ is the scattering angle in radians. The anisotropy factor therefore affected the angular distribution, and so the amount of forward direction maintained by the photon. g may range between −1 (absolute backward scattering) and 1 (absolute forward scattering).

It is possible that the boundary of a medium may fall closer to the photon along its intended unit direction vector of travel, than the scattering point it was intending for. In this instance the packet traveled only as far as the boundary where its new direction was determined by either reflection or refraction, as decided using Fresnel equations for unpolarized light to determine the probability of reflection, and a uniform random deviate to sample. In the condition that the packet refracts, and enters into a new medium, the optical properties as observed by the photon, were updated to reflect the new surrounding environment.

The photon will continue to propagate and scatter until it exits the domain of the simulation at which point its remaining statistical weight was re-added to the remaining pool to be simulated. In the event that a packet’s statistical weight was reduced such that the computational burden of simulating it outweighs its statistically significant contribution to the simulation, the packet underwent a roulette scheme.

The simulation observed which domain cell a photon packet is within at all times, though this does not affect the photon’s journey in any way. In doing so the simulation could observe and record which photo-events occurred in which cells, and so could build up a statistical event map of the domain.

As a photon packet propagates a distance l through a cell, it contributes to the energy density U in that cell by an amount $\frac{\varepsilon \delta t}{\Delta t}$ where $\delta t=\frac{n}{c}\cdot l$ where n is the refractive index of the medium. By performing this propagation procedure each time one of the large number of photons passes through the cell, over the simulation an estimate of U in each cell is gained for each cell within the domain. That is, for a given cell of volume V being traversed by photon packets tracing paths of length l, the energy density is:(7)$$U=\frac{1}{V}\cdot \frac{\varepsilon }{\Delta t}\frac{n}{c}\sum^{\;}l$$

Furthermore since
(8)$$U=4\pi \cdot J\frac{n}{c}$$
and the absorbance rate is
(9)$$\dot{A}=4\pi \cdot J\cdot \mu_{a}$$
we see that
(10)$$A=\frac{1}{V}\cdot \frac{\varepsilon }{\Delta t}\sum^{\;}l\cdot \mu_{a}.$$

We could therefore record the photo-power absorbance of each domain cell. This absorbed power could then be converted into a corresponding thermal effect and the resulting time dependent diffusion could be subsequently calculated using a diffusive initial value problem finite difference method.

The absorption coefficient, *μ*_a_, and scattering coefficient, *μ*_s_, are the two dominant properties of each material which determine how light moves through tissue. The values used for the simulation are shown in Table 1.

These values have been experimentally determined: Jaques [17] has performed a comprehensive investigation into light propagation through tissue, which is summarized for normal tissue and tumour tissue by Campbell et al. [18]; while gold nanorod values can be found in work by He et al. [19] and Jain et al. [20]. In our simulations with GNR infused tumour tissue, the *μ*_a_ and *μ*_s_, are the sum of the tumour tissue values and the gold nanorod values. The *μ*_a_ and *μ*_s_ values are related to the concentration of the material, which is particularly relevant to GNRs as the initial injected concentration will decrease over time through blood perfusion and clearance.

Dickerson et al. used GNRs that were functionalised with a biocompatible molecule polyethylene glycol (PEG). The impact of the PEG functionalisation on the scattering and absorption coefficients of the gold nanorods is likely to be vanishingly small. This is because PEG does not efficiently absorb light at 800 nm [21], unlike gold nanorods (with an aspect ratio of 4). To illustrate this point, the same weight of gold nanorods (~1 mg/mL) has an OD = ~50 at 800 nm regardless of whether it is citrate stabilised (i.e., bare) or PEG-functionalised (see nanocomposix.com products for examples). If the PEG made a substantial difference to absorbance, one would expect the OD at 800 nm to be much higher for the PEG-functionalised GNRs for the same weight of gold, as the PEG would be absorbing a significant fraction of the light and increasing the OD of the solution as a whole.

In the experimental data from Dickerson et al. [22] that we simulated, they injected a concentration of optical density OD_800nm_ = 40 into the tumours and then left to equilibrate for two minutes before irradiation commenced. As the actual concentration of GNRs in the tumours were not known, we ran a number of simulations and found that *μ*_a_ = 12 cm^−1^ and a *μ*_s_ = 1.2 cm^−1^ were required to fit the heat rise data (note the ratio of absorption to scattering does not change regardless of the concentration).

To simulate heat diffusion we used Penne’s Bioheat transfer equation [23], implemented in MATLAB through the package ‘k-wave’ (www.k-wave.org), using the ‘kWaveDiffusion’ class which is given by:(11)$$\rho c\frac{\partial T}{\partial t}\;=\;\nabla \left(k\nabla T\right)-B\left(T-T_{a}\right)+Q\;\;$$
where the terms, symbols and values are summarized in Table 2. Note in Equation (12), *B* = blood specific heat capacity × blood perfusion rate × blood density.

All visualization, data analysis and figure preparation for this manuscript was implemented in MATLAB. All code used can be found at www.github.com/charliejeynes/photothermal, including the raw datacubes produced in ARCTORUS.

## 3. Results

Figure 1 shows a schematic of the skin cancer model dimensions used within the simulations, which are based on stochastic Monte Carlo trajectories of photons as they pass through various media (see Methods for details). Briefly, the simulations are constructed by first specifying a list of optical media, and then subsequently binding these materials within ‘smooth’ trimesh surfaces. This both allows photons to accurately refract and reflect across arbitrarily shaped surfaces, as well as providing a means for photons to derive their current optical parameters. The domain is further resolved using a three-dimension (3D) Cartesian grid which is used to calculate Monte Carlo estimators for the absorption rate.

Within the simulation, an oblate spheroid (height 0.5 mm, diameter 1 mm), representative of a tumour, was embedded at 1 mm depth within a cuboid of normal tissue (see Figure 1). The size of the cuboid composed of normal tissue was great enough such that photons did not escape from any of the sides nor the base. The optical media forming the tumour could be modified to match those of either air, normal tissue, tumour tissue or tumour tissue infused with Gold NanoRods (GNRs). The absorbance and scattering values for the various tissue types and GNRs used are summarized in Table 1. For the simulations, a 1W light source with a radius of 3 mm, illuminated the model so that most photons first travel through air, then normal tissue before entering the tumour (a fraction of photons will scatter missing the tumour entirely). Each simulation followed the trajectories of 10^9^ photons within the model, resulting in an energy density datacube (W/m^3^), for each 0.1 mm^3^ grid cell voxel.

Figure 2 shows a comparison between simulations from a skin tumour model where the tumour has been infused with light absorbing gold nanorods, compared to the control simulation without nanorods. Specifically, Figure 2 shows a 2D cross-section of the 3D datacube, where the output is in W/m^3^. Here it can be seen that the nanorods significantly increased the energy deposited within the tumour compared to the control. The line profile shows that the top of the control tumour received ~0.5 × 10^7^ W/m^3^ compared to ~3 × 10^7^ W/m^3^ for the nanorod infused tumour.

The output datacube (W/m^3^) from the simulations is then passed into a “Bioheat Transfer” function, based on Penne’s biological heat transfer equation [23] (see Methods). This takes the energy deposited per voxel (W/m^3^) and calculates heat diffusion within tissues over time, considering variables such as specific heat capacity, medium diffusivity and blood perfusion. The resulting change in temperature per second is shown in Figure 3. Figure 3, top panel, shows heat diffusion after one second, while Figure 3, bottom panel, is after 600 s (10 min). Note that over time there was constant illumination from the 1 W light source. After one second of illumination and diffusion, the temperature inside the tumour had risen by ~0.7 °C for the control tumour, and ~4 °C for the nanorod infused tumour. As expected, that heat diffusion smoothed out the energy deposited such that the difference in the heat rise between the top and bottom of the tumour was less compared to the energy deposited. Figure 3, bottom panel, shows that after 10 min, the heat rise was approaching 8 °C for the control tumour and 25 °C for the nanorod infused tumour.

Figure 4 shows heat rise over time comparing our simulations to data from Dickerson et al. [22]. In their experiment, tumours in mice were directly injected with gold nanorods (at a concentration of optical density (OD) = 40) with subsequent illumination from an 800 nm laser (1 W/cm^2^, 6 mm spot size) for 10 min. The temperature inside the tumour was measured every 20 s with an inserted micro-temperature probe.

Dickerson et al. injected gold nanorods that were ~12 nm in width and ~50 nm in length corresponding to an aspect ratio of 4—as these have a peak plasmonic resonance at 800 nm. For our model we needed to know the absorption and scattering coefficients of these gold nanoparticles (GNPs), which are not fixed values but change proportionally with the concentration. Fortunately, He et al. experimentally showed that at 800 nm the ratio between the maximum scattering coefficient and the maximum total extinction coefficient (i.e., absorption + scattering) for these types of GNRs (aspect ratio 4), is 0.1. Interestingly, Dickerson et al. did not measure the final concentration of gold nanorods within the tumours, rather they just reported the injected concentration. Therefore, we simulated a range of absorption and scattering coefficients (with a fixed ratio of 0.1) to obtain values that most closely fitted Dickerson’s heat rise data (see Methods for more details). These values are shown in Table 2. We found approximately a four-fold reduction in the GNR concentration comparing our simulated absorption and scattering values to their injected OD = 40 value. This seems reasonable given that there was a two-minute interval between injection and the start of irradiation, and in that time, nanorods would have been cleared from the tissue through blood perfusion, with a fraction (by our calculations ~25%) of nanorods becoming trapped in the tumour.

Figure 4 shows reasonable agreement between their data and our simulated results, where there was an initial steep rise in temperature in the first three minutes of irradiation followed by a plateau. For the control tumours, the temperature rise never exceeded more than 8 °C, whereas for the tumours containing gold nanorods the temperature rose rapidly to exceed 20 °C. The fit between the mean experimental values and the simulations was between 5–30%. This was well within the uncertainties associated with the absorbance, *μ*_a_, and scattering, *μ*_s_, values used in the simulations (more on this in the discussion). Most of the simulation data also fell within the standard deviation (~20%) of the experimental data.

Dickerson et al. also showed the nanorod-NIR treatment prevented tumour growth, while the control tumours (with no injected nanorods) tripled in volume over two weeks after irradiation. This matches well with our simulation shown in Figure 5 which profiles the estimated cell death throughout the normal tissue and the tumour. We estimated cell death using cumulative equivalent minutes at 43 °C (CEM43). As cell death is a function of both temperature and time held at that temperature, CEM43 has become a standard method to normalise data for easy comparison to cell survival profiles [24].

Figure 5, top panel, shows CEM43 [24,25], which is calculated as Equation (12):CEM 43 °C = *tR*^(43 − *T*)^(12)
where CEM 43 °C = cumulative number of equivalent minutes at 43 °C, *t* = time interval (min), *T* = average temperature during time interval *t*, *R* is the number of minutes needed to compensate for a one-degree temperature change either above or below the threshold. The resultant CEM 43 °C value represents the entire history of the exposure.

Figure 5, bottom panel, shows the estimated cell death profile from our simulations based on fitted cell survival curves. These survival curves were obtained from human cancer cell line kept at 43 °C over time and showed an exponential decay in survival with about 10% cells surviving after 2 h [24].

Figure 6 shows an analysis of the damage to surrounding normal tissue, using the estimated cell based on CEM43. Here, we have masked out the tumour region to easily visualize the boundary between the tumour and normal tissue. The line profile shows that ~1 mm around the tumour received a thermal dose that will kill close to 100% of the normal cells. This ‘collateral’ damage of the treatment is actually relatively difficult to test for experimentally, and so this estimation through our simulation is a very useful tool to quickly visualize likely damage to normal tissue that the treatment will cause.

## 4. Discussion

Photothermal therapy using nanoparticles has an extensive experimental history, with researchers using a variety of nanoparticles and irradiation protocols (reviewed in [1]). In our work, the aim was to develop and validate a theoretical approach to thermal dose deposition within tissue, so that: (i) the spatial and temporal distributions of thermal dose within the tissue could be analysed —which is technically unfeasible with most experimental set-ups; (ii) offer a predictive tool which allows researchers to estimate likely thermal effects of concentrations and spatial profiles of nanoparticles as well as irradiation dose that could influence tumour treatment efficacy; and (iii) give an estimate of the concentration of gold nanorods that is needed to give a heat rise that results in adequate cell kill within the tumour (many experimental reports do not measure the concentration of GNRs in the tumour at the point of irradiation—rather they only know the injected concentration). Another factor that could effectively lower the concentration of nanorods is a sintering effect where localized heat deforms the nanoparticles such that they no longer exhibit plasmonic absorption at the irradiation wavelength [26].

Our Monte Carlo radiative transfer code builds on work originally designed to model light transfer in astrophysical systems [16]. Whilst the scale and composition of the system within the simulation changes drastically between the astrophysical and the biophysical domain, the principles of radiative transfer remain the same, with only surface interactions requiring special attention. As such, the principles of the TORUS [16] code have been applied to a new radiative transfer engine designed to also handle surfaces making it suitable for accurate simulation of biological systems. There are similar existing models and codes that have been developed, most notably to simulate photodynamic therapy using the drug PpIX [13,14]. Uniquely, however, our new simulation code—so called ARCTORUS—allows the import of complex objects such as realistic tumour geometries and blood vessel capillary networks. The significance here is that the complexity of the geometry and optical properties of these objects could very well make a large difference to light penetration and will be the subject of future investigation by our group.

Surprisingly, given the large amount of experimental endeavor into nanoparticle photothermal therapy, there has been relatively little theoretical work simulating thermal dose from nanoparticles using NIR light sources.

One notable example by Cuplov et al. [15] showed that using GATE, a Monte Carlo toolkit based on the open source framework GEANT4, they could accurately model the heat rise from experimental data obtained from Hirst et al. [27]. Their work is the first example in this field of coupling a Monte Carlo code which accurately accounts for photon absorbance, scattering and refraction in complex medium with an analytical code that solves Penne’s bioheat equation [23]. Our work employs a similar approach, but with our Monte Carlo code we have the flexibility of adding complex objects to account for more realistic tumour shapes and sizes as well as skin features such as hair follicles, sweat glands and blood network embedded in fatty tissue. As for the bioheat transfer algorithm, we utilized ‘k-wave’, an experimentally validated freely available code implemented in MATLAB.

Our analysis specifically focused on cell death within the tumour and the surrounding normal tissue region—whereas Cuprov et al. did not include any spatial distributions of dose or prediction of cell death. Their work focused on the errors associated with varying the thermal diffusivity rate which can change for different types of tissue, but found that it made less than <5% difference to thermal rise regardless which constant was chosen. On this note, it is interesting that the absorbance and scattering values used for the Monte Carlo photon trajectories, and the variables for heat transfer such as thermal diffusivity rate, are estimates of the real value, but nevertheless appear to give reasonable fits to the experimental results (see Figure 3). These estimates based on experimental measurements and fitting (see a full review in Jacques [17], and more specifically for tumour and normal tissue see Campbell et al. [18]) have relatively large (~20%) uncertainties. This means that the relative difference between the mean of the experimental data and our simulation of between 5–30% is well within the limits of the uncertainties.

Clinically, it is of paramount importance to estimate the percentage cell death spatially within a tumour as any cell survival can lead to regrowth and/or secondary tumours. Further, the damage to surrounding normal tissue must be known as the tumor could be adjacent to vital structures where damage could lead to side effects/complications. Admittedly, this is less important in skin cancers, although collagen damage could result in unsightly scarring—but if photothermal therapy is to be used for other cancer applications, such as head and neck malignancies—avoiding damage to normal tissue will become increasingly important. Our simulations allow for spatial analysis of thermal rise in tissue, and by using the CEM43 index, estimates cell kill within regions. Future work will involve simulating more complex tumour geometries, more closely representing tumours found in the clinic such as nodular basal cell carcinoma, to optimise irradiation and nanoparticle dose regimes, which would provide effective treatment outcomes for these cancers. Photothermal therapy rather than photodynamic therapy could also provide better clinical outcomes in tumours where poor oxygenation is thought to be a likely cause for poor treatment response.

## 5. Conclusions

Our work and the possible implications can be summarised as follows:

(i) Our approach is to create a ‘virtual laboratory’ where we can test hypothesis related to light interaction and dose in tissue. We have used nanoparticle photothermal therapy as a case study of how the code can be used and be validated against experimental results. We have other projects extending the applications into other fields of cancer therapy such as photodynamic therapy; (ii) Our results complement experimental data and can retrospectively analyse the dynamics of dose distribution in tissue which is information not easily obtained with many experimental set-ups; (iii) We envisage our approach being used as a predictive tool to enable clinicians to plan treatment based on quantitative outputs from our model. Specifically, in the case of possible nanoparticle photothermal therapy, this would include concentration of nanoparticle needed in the tumour for cell killing, and dose of NIR illumination to induce critical heat rise; (iv) We have adapted an astrophysical code to the biophysical domain to investigate questions of light transport in tissue. This is of interest to astrophysicists as it is an impact case study of the utility of astrophysics tools in other fields.

## Figures and Tables

**Figure 1 biomolecules-09-00343-f001:**
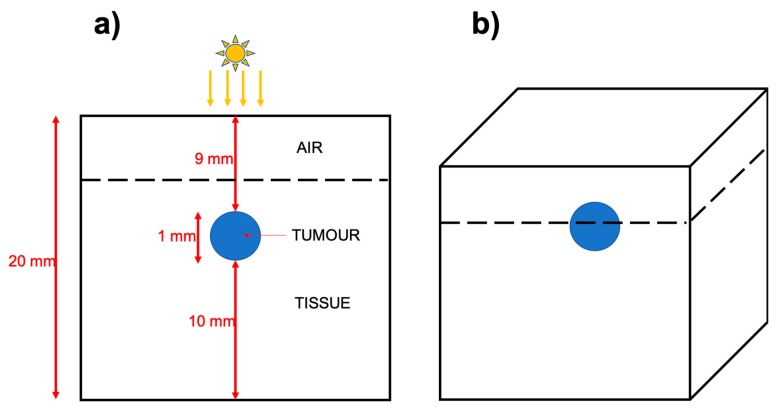
Schematic showing the dimensions of the simplified skin cancer model. (**a**) A two-dimension (2D) slice through the simulated world, which is a cartesian grid illustrated in (**b**). The light source is 1 W and has a radius of 3 mm and is set to irradiate the top and centre of the model. In the simulations, we compare a control, to a tumour infused with gold nanorods, with the absorbance and scattering values (respectively *μ*_a_ and *μ*_a_) shown in Table 1.

**Figure 2 biomolecules-09-00343-f002:**
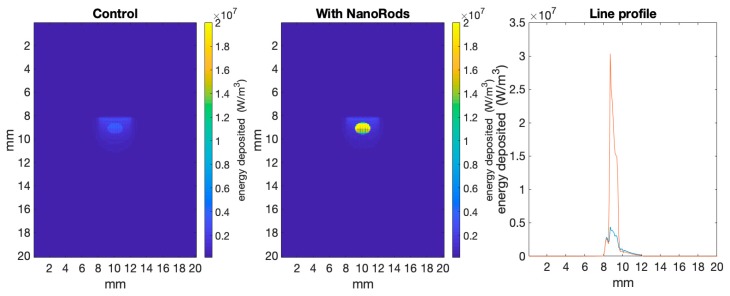
Shows energy deposited per voxel in a 2D slice through the simulations from ARCTORUS, comparing the control to tumour tissue infused with gold nanorods. The line profile bisects the model from top to bottom at 10 mm. The scattering and absorbance values used in the simulations are shown in Table 1.

**Figure 3 biomolecules-09-00343-f003:**
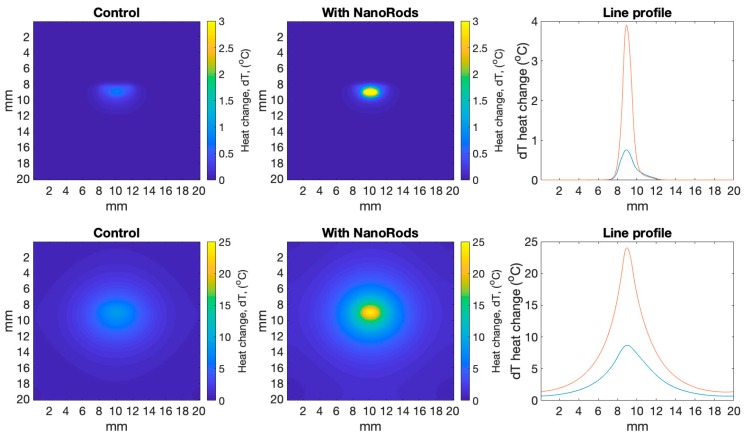
Shows heat rise (°C) per voxel in a 2D slice through the simulations from ARCTORUS comparing the control with tissue infused with gold nanorods, after one second (top panel) and 600 s (bottom panel). The line profile bisects the model from top to bottom at 10 mm.

**Figure 4 biomolecules-09-00343-f004:**
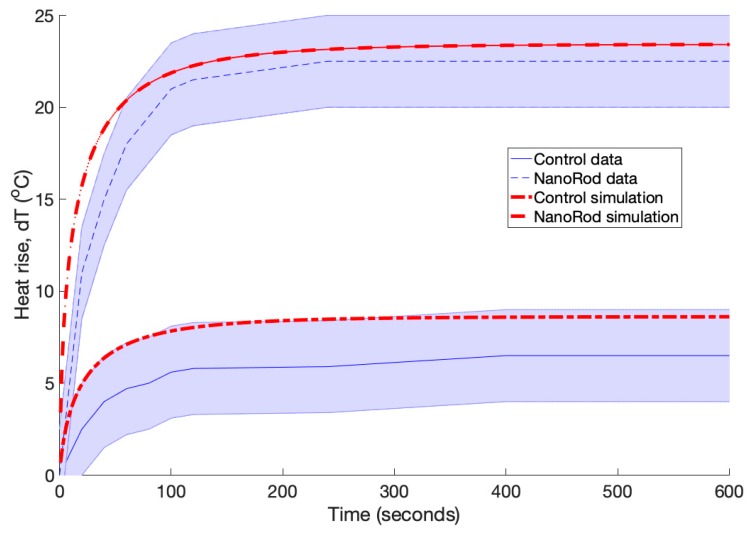
Shows heat rise (°C) in the centre of tumour over 10 min (600 s) comparing simulations to experimental data taken from Dickerson et al. [22]. In their experiment, treatments were performed by administration of 15 µL of pegylated gold nanorods (OD_800_ = 40, 2 min accumulation) followed by 10 min of 0.9–1.1 W NIR laser exposure. Sham/NIR treatments were performed by administration of 15 µL 10 mM PBS and NIR laser exposure of comparable time and power density.

**Figure 5 biomolecules-09-00343-f005:**
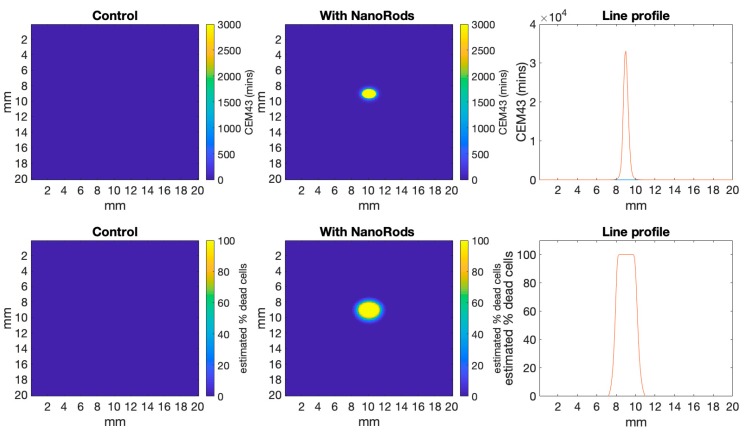
Shows the heat diffusion simulation at 10 min (600 s) converted to cumulative equivalent minutes at 43 °C (CEM43) (Top Panel), and the estimated percentage cell death that is the result of the time spent at CEM43 (Bottom Panel). This shows that within the tumour region 100% of cells are likely to die due to the thermal dose they receive, with this likelihood dropping off steeply with distance from the tumour (more details in Figure 6).

**Figure 6 biomolecules-09-00343-f006:**
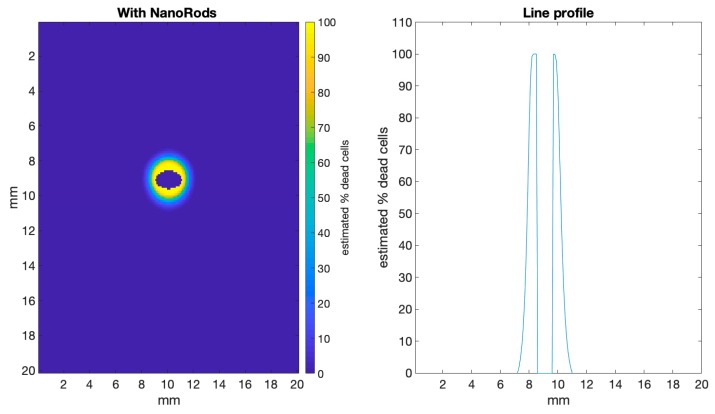
Shows the same data as in Figure 5 (bottom panel) but with the tumour region masked, so that the damage to normal tissue can be more easily visualized. The line profile, with the tumour data removed, clearly shows that normal tissue ~1 mm around the tumour receives a dose where close to 100% of the cells are estimated to die.

**Table 1 biomolecules-09-00343-t001:** Absorbance, *μ*_a_, and Scattering, *μ*_s_, values used in the simulations.

Material	Absorption Coefficient (cm^−1^), *μ*_a_,	Scattering Coefficient (cm^−1^), *μ*_s_
Normal tissue	0.7	36.7
Tumour tissue	2.3	21.2
Gold Nanorods (GNRs)	12	1.2
Tumour tissue with GNRs	14.3	22.4

**Table 2 biomolecules-09-00343-t002:** Values used for the Bioheat Transfer Equation.

Term	Symbol	Value
Tissue Density, (kg/m^3^)	*ρ*	1079
Thermal conductivity, (W/m·K)	*k*	0.53
Tissue specific heat capacity (J/(kg K)	*c*	3540
Blood Density (kg/m^3^)	*B*	1060
Blood specific heat capacity (J/(kg K)	*B*	3617
Blood perfusion rate (1/s)	*B*	0.01
Blood ambient temperature (°C)	*T_a_*	37
Local Temperature (°C)	*T*	Initially 37
Volume rate of heat deposition, W/m^3^	*Q*	ARCTORUS data

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
