# Peer review of "Monte Carlo Simulations of Heat Deposition during Photothermal Skin Cancer Therapy Using Nanoparticles"

_biomolecules, 2019, doi:10.3390/biom9080343_

Round 1
Reviewer 1 Report
The present manuscript describes a study on the photothermal therapy activated by nanoparticles from a purely theoretical perspective. Monte Carlo has been used as the main calculation procedure from a well-known expert in the field. The results are sound and the manuscript reads well. The provided insights might be relevant for both theoretical and experimental researchers working in the field of photothermal therapy for cancer therapy. The significance of the study for assessing the actual mechanism of photothermal therapy in human tissues is remarkable.
I certainly recommend the manuscript for publication in Biomolecules, with only two minor comments:
* Abstract and introduction sections do not mention plasmonic nanoparticles, although according to the results and setup of the study, I would assume plasmonic gold nanorods are the object of the study. Please refer to this in abstract and introduction.
* “This seems reasonable given that there was a two‐minute interval between injection and the start of irradiation, and in that time, nanorods would have been cleared from the tissue through blood perfusion, with a fraction (by our calculations ~25%) of nanorods becoming trapped in the tumour.” This process seems plausible. Indeed, a sintering of the gold nanorods might also be expected after the plasmonic-induced heating, effectively leading to a decrease on the concentration of gold nanorods.
Author Response
Reviewer 1
The work by Jeynes et al. described so far Monte Carlo simulations on heat transfer during phototermal therapy. The work is interesting: However, it remains simply as a descriptive work and there is no discussion about the possible implications of the obtained results.
Reply: The possible implications of our obtained results can be summarised as follows:
i) Our approach is to create a ‘virtual laboratory’ where we can test hypothesis related to light interaction and dose in tissue. We have use nanoparticle photothermal therapy as a case study of how the code can be used and be validated against experimental results. We have other projects extending the applications into other fields of cancer therapy such as photodynamic therapy; ii) Our results complement experimental data and can retrospectively analyse the dynamics of dose distribution in tissue which is information not easily obtained with many experimental set-ups; iii) We envisage our approach being used as a predictive tool to enable clinicians to plan treatment based on quantitative outputs from our model. Specifically, in the case of possible nanoparticle photothermal therapy, this would include concentration of nanoparticle needed in the tumour for cell killing, and dose of NIR illumination to induce critical heat rise; iv) We have adapted an astrophysical code to the biophysical domain to investigate questions of light transport in tissue. This is of interest to astrophysicists as it is an impact case study of the utility of astrophysics tools in other fields;
The above comments have now been included in a conclusion section.
Furthermore, the work is submitted to a journal entitled biomolecules and there are no biomolecules in the manuscript, authors dealts with particles and tissues.
Reply: We were invited to submit this article to this special addition of Biomoleculesentitled “Nanoparticles in cancer therapy”. We have assurance from the editor that our work is within the scope of the journal in this instance.
In summary, even the work could present impact in the development of new therapies against cancer, the presentation of the results in the manuscript is rather poor and the topic is out of the scope of the journal.
Reply: Please could the reviewer expand on the comment that the presentation is rather poor. We have endeavoured to capture the essence of the simulations in the visualisations and quantitative line plots as best we could. We apologies if these are not up to standard – but if at all possible, could we have some specifics on how to improve them.
Reviewer 2 Report
The manuscript written by Jeynes et al. presents a new custom-written Monte Carlo model to simulate the temperature rise profile of gold Nanorod infused skin cancer model during a NIR photodynamic treatment. The presented piece of research aims to demonstrate the thermal dose deposition within a model tissue throughout the thermal treatment. The model is particularly based on the simulation of absorbance, scattering, refraction, and reflection of individual photon packets though optical media. Therefore, the simulations were run based on the experimentally determined absorption/scattering coefficient data of tissue and nanoparticles in literature. The work has been carefully carried out by considering the actual amounts of gold nanorods retained within tumor tissue rather than the injected initial amounts; as well as the damage caused to healthy tissue during the photodynamic treatment.
Although I am not capable of discussing the advantages/disadvantages of the Monte Carlo model employed in this study, I would like to raise following questions regarding the work:
- In order to specify the concentration induced effects of the gold nanorods in the tumor tissue, authors used optical density values, absorption/scattering coefficients of the gold nanorods, which were gathered from literature. By limiting the simulations to these values, size and shaped dependent absorption/scattering properties of gold nanorods are somehow ignored. However, as the authors also referenced, it is known that the spectral values of the gold nanorods are very much dependent on the aspect ratio of the particles [ J. Phys. Chem. C, Vol. 114, No. 7, 2010 and J. Phys. Chem. B, Vol. 110, No. 14, 2006 ]. In my opinion, it is crucial to provide information about the size or the aspect ratio of the nanorods used in the simulations. Authors should at least specify what was the size (aspect ratio) of the simulated gold nanorods that corresponds to given absorption/scattering data.
- As far as I see from the previous publication of the Authors, where they detailed the Monte Carlo model they used [Astronomy and Computing 27 (2019) 63–95)], the model is capable of defining the “grain sizes”. Is there a specific reason why authors did not provide any information about the nanorod sizes simulated in the model ?
- Authors compared the results of simulations to the data acquired from Dickerson et al..However, the gold nanorods reported Dickerson et al. were capped with PEG-SH molecules. Did the simulations take the presence of the PEG-SH molecules into account? Were the absorption/scattering coefficients used in simulations suitable for PEG-SH capped gold nanorods ?
I believe that the answers of these questions are very important for the potential readers working on nanomedical applications of gold nanoparticles. Providing further information on these questions would absolutely improve the quality of the manuscript.
Author Response
Reviewer 2
The present manuscript describes a study on the photothermal therapy activated by nanoparticles from a purely theoretical perspective. Monte Carlo has been used as the main calculation procedure from a well-known expert in the field. The results are sound and the manuscript reads well. The provided insights might be relevant for both theoretical and experimental researchers working in the field of photothermal therapy for cancer therapy. The significance of the study for assessing the actual mechanism of photothermal therapy in human tissues is remarkable.
I certainly recommend the manuscript for publication in Biomolecules, with only two minor comments:
* Abstract and introduction sections do not mention plasmonic nanoparticles, although according to the results and setup of the study, I would assume plasmonic gold nanorods are the object of the study. Please refer to this in abstract and introduction.
Reply: Plasmonic gold nanoparticles are indeed the object of study. We have now referred to this in the abstract and introduction.
In the Abstract:“Photothermal therapy using nanoparticles is a promising new approach for the treatment of cancer. The principle is to utilise plasmonic nanoparticle light interaction for efficient heat conversion. However, there are many hurdles….”
In the introduction:“Gold nanorods can be designed with an aspect ratio that is highly absorbing in the near infra-red light region due to a surface plasmon resonance effect [2].”
* “This seems reasonable given that there was a two‐minute interval between injection and the start of irradiation, and in that time, nanorods would have been cleared from the tissue through blood perfusion, with a fraction (by our calculations ~25%) of nanorods becoming trapped in the tumour.” This process seems plausible. Indeed, a sintering of the gold nanorods might also be expected after the plasmonic-induced heating, effectively leading to a decrease on the concentration of gold nanorods.
Reply: Sintering of the GNRs is also possible, effectively leading to a decrease in the concentration of GNR – and we have now commented on this in the discussion and included a reference to this possible mechanism (Harris-Birtillet al. 2017)
“Another factor that could effectively lower the concentration of nanorods is a ‘sintering’ effect where localized heat deforms the nanoparticles such that they no longer exhibit plasmonic absorption at the irradiation wavelength [20].”
Reviewer 3 Report
The work by Jeynes et al. described so far Monte Carlo simulations on heat transfer during phototermal therapy. The work is interesting: However, it remains simply as a descriptive work and there is no discussion about the possible implications of the obtained results. Furthermore, the work is submitted to a journal entitled biomolecules and there are no biomolecules in the manuscript, authors dealts with particles and tissues.
In summary, even the work could present impact in the development of new therapies against cancer, the presentation of the results in the manuscript is rather poor and the topic is out of the scope of the journal.
Author Response
Reviewer 3
The manuscript written by Jeynes et al. presents a new custom-written Monte Carlo model to simulate the temperature rise profile of gold Nanorod infused skin cancer model during a NIR photodynamic treatment. The presented piece of research aims to demonstrate the thermal dose deposition within a model tissue throughout the thermal treatment. The model is particularly based on the simulation of absorbance, scattering, refraction, and reflection of individual photon packets though optical media. Therefore, the simulations were run based on the experimentally determined absorption/scattering coefficient data of tissue and nanoparticles in literature. The work has been carefully carried out by considering the actual amounts of gold nanorods retained within tumor tissue rather than the injected initial amounts; as well as the damage caused to healthy tissue during the photodynamic treatment.
Although I am not capable of discussing the advantages/disadvantages of the Monte Carlo model employed in this study, I would like to raise following questions regarding the work:
- In order to specify the concentration induced effects of the gold nanorods in the tumor tissue, authors used optical density values, absorption/scattering coefficients of the gold nanorods, which were gathered from literature. By limiting the simulations to these values, size and shaped dependent absorption/scattering properties of gold nanorods are somehow ignored. However, as the authors also referenced, it is known that the spectral values of the gold nanorods are very much dependent on the aspect ratio of the particles [ J. Phys. Chem. C, Vol. 114, No. 7, 2010 and J. Phys. Chem. B, Vol. 110, No. 14, 2006 ]. In my opinion, it is crucial to provide information about the size or the aspect ratio of the nanorods used in the simulations. Authors should at least specify what was the size (aspect ratio) of the simulated gold nanorods that corresponds to given absorption/scattering data.
Reply: This is a very valid point and we thank the reviewer for highlighting it. In the paper, we never actually mentioned the size or aspect ratio of the gold nanorods we were simulating. We have now corrected this by stating explicitly in the Results:
“Dickerson et al. injected gold nanorods that were ~12 nm in width and ~50 nm in length corresponding to an aspect ratio of 4 – as these have a peak plasmonic resonance at 800nm. For our model we need to know the absorption and scattering coefficients of these GNPs, which are not fixed values but change proportionally with the concentration. Fortunately, He et al.[25]experimentallyshowed that at 800 nm the ratio between the maximum scatteringcoefficient and the maximum totalextinction coefficient (i.e. absorption+scattering) for these types of GNRs (aspect ratio 4), is 0.1. Interestingly, Dickerson et al. did not measure the finalconcentration of gold nanorods within the tumours, rather they just reported the injectedconcentration. Therefore, we simulated a range of absorption and scattering coefficients (with a fixed ratio of 0.1) to obtain values that most closely fitted Dickerson’s heat rise data (see Methods for more details). These values are shown in Table 2. We found approximately a 4-fold reduction in the GNR concentration comparing our simulated absorption and scattering values to their injected OD=40 value. This seems reasonable given that there was a two-minute interval between injection and the start of irradiation, and in that time, nanorods would have presumably cleared from the tissue through blood perfusion, with a fraction (by our calculations ~25%) of nanorods becoming trapped in the tumour.”
- As far as I see from the previous publication of the Authors, where they detailed the Monte Carlo model they used [Astronomy and Computing 27 (2019) 63–95)], the model is capable of defining the “grain sizes”. Is there a specific reason why authors did not provide any information about the nanorod sizes simulated in the model?
Reply: No there is no specific reason we did not provide information as to the sizes of the nanorods simulated. We have now corrected this by including in the paper that we are simulating GNRs with a ~12nm width and ~50 length (aspect ratio 4), as explained in the comment above.
- Authors compared the results of simulations to the data acquired from Dickerson et al..However, the gold nanorods reported Dickerson et al. were capped with PEG-SH molecules. Did the simulations take the presence of the PEG-SH molecules into account? Were the absorption/scattering coefficients used in simulations suitable for PEG-SH capped gold nanorods ?
Reply: This is a good point and one which we now comment on in the Methods.
“Dickerson et al.used GNRs that were functionalised with a biocompatible molecule polyethylene glycol (PEG). The impact of the PEG functionalisation on the scattering and absorption coefficients of the gold nanorods is likely to be vanishingly small. This is because PEG does not efficiently absorb light at 800nm (Jia et al. 2009), unlike gold nanorods (with an aspect ratio of 4). To illustrate this point, the same weight of gold nanorods (~1mg/ml) has an OD=~50 at 800nm regardless of whether they are citrate stabilised (i.e. bare) or PEG-functionalised (see nanocomposix.com products for examples). If the PEG made a substantial difference to absorbance, one would expect the OD at 800nm to be much higher for the PEG-functionalised GNRs for the same weight of gold, as the PEG would be absorbing a significant fraction of the light and increasing the OD of the solution as a whole.”
I believe that the answers of these questions are very important for the potential readers working on nanomedical applications of gold nanoparticles. Providing further information on these questions would absolutely improve the quality of the manuscript.
Reviewer 4 Report
Integrating with nanoparticles, photothermal therapy is a potential way to treat cancer. However, numerous challenges remain for this approach. In this manuscript, by adopting Monte Carlo simulations, the authors investigated the spatial thermal dose in a skin cancer model. They proposed the concentration of gold nanorods for a heat rise. They also provided some results that cannot be obtained by experiments (e.g. the distributions of thermal dose around tumor and its surround tissue). By employing the cumulative equivalent minutes at 43°C, they examined the percentage cell kill across tumor and its surrounding normal tissue. It is an interesting manuscript. However, I have some concerns.
Please provide conclusion section in the text.
To better reproduce the simulations in this manuscript, a more detailed modeling is needed.
Page 7 of 14 “Figure 5, top panel, shows CEM43, which is calculated as….” If possible, please provide reference(s) for this equation.
Page 9 of 14 “There are similar existing models and codes that have been developed”. Reference(s) are needed to support this statement.
Page 11 of 14, line 339: If possible, please provide reference(s) for this equation
Author Response
Reviewer 4
Integrating with nanoparticles, photothermal therapy is a potential way to treat cancer. However, numerous challenges remain for this approach. In this manuscript, by adopting Monte Carlo simulations, the authors investigated the spatial thermal dose in a skin cancer model. They proposed the concentration of gold nanorods for a heat rise. They also provided some results that cannot be obtained by experiments (e.g. the distributions of thermal dose around tumor and its surround tissue). By employing the cumulative equivalent minutes at 43°C, they examined the percentage cell kill across tumor and its surrounding normal tissue. It is an interesting manuscript. However, I have some concerns.
Please provide conclusion section in the text.
Reply: We have now provided a conclusion paragraph. See the first response to Reviewer 1.
To better reproduce the simulations in this manuscript, a more detailed modeling is needed.
Reply: We have now re-written the Methods such that the Monte Carlo simulation is explained in much greater depth (please see the amended manuscript).
Page 7 of 14 “Figure 5, top panel, shows CEM43, which is calculated as….” If possible, please provide reference(s) for this equation.
Reply: We have now given a reference (ref 20) “Sapareto, S.A. and W.C. Dewey, Thermal Dose Determination in Cancer-Therapy.International Journal of Radiation Oncology Biology Physics, 1984. 10(6): p. 787-800.”
Page 9 of 14 “There are similar existing models and codes that have been developed”. Reference(s) are needed to support this statement.
“
Reply: this sentence has been referenced (see refs 13, 14) “There are similar existing models and codes that have been developed [13, 14], most notably to simulate photodynamic therapy using the drug PpIX.”
Page 11 of 14, line 339: If possible, please provide reference(s) for this equation
Reply: This has now been referenced to (ref 16):“Harries, T.J., et al., The TORUS radiation transfer code.Astronomy and Computing, 2019. 27: p. 63-95.”
Round 2
Reviewer 3 Report
I recommend the publication of the manuscript in its present form.
Reviewer 4 Report
The authors have considered all points I raised in my initial report. The manuscript is improved. I recommend the publication of this manuscript